# The Impact of Assisted Reproductive Technology on Umbilical Cord Insertion: Increased Risk of Velamentous Cord Insertion in Singleton Pregnancies Conceived through ICSI

**DOI:** 10.3390/medicina59101715

**Published:** 2023-09-25

**Authors:** Eriko Fukuda, Akihiro Hamuro, Kohei Kitada, Yasushi Kurihara, Mie Tahara, Takuya Misugi, Akemi Nakano, Mami Tamaue, Sae Shinomiya, Hisako Yoshida, Masayasu Koyama, Daisuke Tachibana

**Affiliations:** 1Department of Obstetrics and Gynecology, Graduate School of Medicine, Osaka City University, 1-4-3 Asahimachi, Abeno-ku, Osaka 5454-8585, Japan; t_n8_l@yahoo.co.jp; 2Department of Obstetrics and Gynecology, Graduate School of Medicine, Osaka Metropolitan University, 1-4-3 Asahimachi, Abeno-ku, Osaka 5454-8585, Japan; kafukafu0404@yahoo.co.jp (K.K.); v21555o@omu.ac.jp (Y.K.); mtahara@omu.ac.jp (M.T.); t-misugi@omu.ac.jp (T.M.); akeake@omu.ac.jp (A.N.); masayasukoyama@gmail.com (M.K.); dtachibana@omu.ac.jp (D.T.); 3Women’s Health Care Science, Advanced Care Science Field, Graduate School of Nursing, Osaka Metropolitan University, 1-5-17 Asahimachi, Abeno-ku, Osaka 5454-8585, Japan; tamaue.mami@omu.ac.jp; 4Department of Medical Statistics, Graduate School of Medicine, Osaka Metropolitan University, 1-4-3 Asahimachi, Abeno-ku, Osaka 5454-8585, Japan; si22417z@st.omu.ac.jp (S.S.); hisako.yoshida@omu.ac.jp (H.Y.)

**Keywords:** IVF-ET, ICSI, cord insertion (CI), vasa previa, velamentous insertion, feto-fertility

## Abstract

*Background and Objectives*: Vasa previa (VP) is a significant perinatal complication that can have serious consequences for the fetus/neonate. Velamentous cord insertion (VCI) is a crucial finding in prenatal placental morphology surveillance as it is indicative of comorbid VP. Assisted reproductive technology (ART) has been identified as a risk factor for VCI, so identifying risk factors for VCI in ART could improve VP recognition. This study aims to evaluate the displacement of umbilical cord insertion (CI) from the placental center and to examine the relationship between the modes of conception. *Materials and Methods*: We conducted a retrospective study at the Obstetrics Department of Osaka Metropolitan University Hospital in Japan between May 2020 and June 2022. The study included a total of 1102 patients who delivered after 22 weeks of gestation. They were divided into three groups: spontaneous pregnancy, conventional in vitro fertilization (cIVF), and in vitro fertilization/intracytoplasmic sperm injection (IVF/ICSI). We recorded patient background information, perinatal complications, perinatal outcomes, and a numerical “displacement score”, indicating the degree of separation between umbilical CI and the placental center. *Results*: The displacement score was significantly higher in the cIVF and IVF/ICSI groups compared with the spontaneous conception group. Additionally, the IVF/ICSI group showed a significantly higher displacement score than the cIVF group. *Conclusions*: Our study provides the first evidence that the methods of ART can affect the location of umbilical CI on the placental surface. Furthermore, we found that IVF/ICSI may contribute to greater displacement of CI from the placental center.

## 1. Introduction

Vasa previa (VP) is a rare and serious condition that can result in fetal/neonatal hemorrhage due to the laceration of vulnerable fetal vessels that run through the membrane [1]. The recognition of VP before the rupture of membranes and/or labor onset is required when echographic signs of fetal vessels running near the internal cervical os are found [2]. The diagnosis of VP demands meticulous observation and well-experienced evaluation, and the condition requires careful perinatal management in the subsequent pregnancy course [1,2].

In the prenatal surveillance of the placental morphology, velamentous cord insertion (VCI) is a key finding to presume comorbid VP [3]. Placental types of vasa previa can be classified into three types: vasa previa with VCI type, vasa previa with multilobed or succenturiate placenta type, and vasa previa with vessels branching out from the placental surface and returning to the placental cotyledons. Of these, the VCI type and multilobed or succenturiate placenta type are associated with VCI as abnormal umbilical cord insertion [3]. Assisted reproductive technology (ART) has been reported to be one of the risk factors for VCI, although its mechanism still remains unclear [4,5]. In their 1990 paper, Jauniaux et al. first reported the increased risks of abnormal placental shapes and abnormal cord insertions (CIs) into placentas conceived by conventional in vitro fertilization (cIVF) [6]. However, despite the tremendous advantages of ART, such as oocyte/blastocyst vitrification and intra-cytoplasmic insemination, revised validation of the incidence of VCI after ART has never been attempted since the 1990s.

The identification of risk factors of VCI in ART conceptions may offer opportunities for improved recognition of VP. The current study aimed to evaluate the deviations of umbilical CI from the center of the placenta using our original formula and to depict the association of cIVF and IVF/ICSI with VCI.

## 2. Materials and Methods

The medical records of women with a singleton pregnancy who delivered at Osaka Metropolitan University Hospital after 22 weeks of gestation between May 2020 and June 2022 were retrospectively reviewed. All patients gave their informed written consent, and the study protocol was approved by the Institutional Review Board (approval number: 2022-107, 8 October 2022). The women studied were divided into three groups: the spontaneous conception group, the cIVF group, and the IVF/ICSI group. Intrauterine fetal death, multiple pregnancies, unknown method of conception, multiple-lobed and/or accessory placentas, undelivered placentas due to adhesion, and incomplete entry were excluded from the study. We also excluded the cases conceived by fresh embryo transfer since frozen embryo transfer is often chosen to improve pregnancy rates.

After the delivery of the placenta, midwives and/or obstetricians examined the placental morphology and recorded the long/short diameters of the placenta and the longest/shortest distance from CI to the margin. Figure 1 shows how the umbilical CI was measured. A and B indicate the longest and shortest diameters of the placenta, respectively. C indicates the longest distance from CI to the margin of the placenta, and D indicates the shortest distance from CI to the margin of the placenta. In cases of VCI, D was measured as a negative value. Since the placenta is not perfectly round, it is difficult to assess the off-centering of the umbilical CI. In this study, we defined the deviation from the center as the “displacement score” and devised the following formula to evaluate it: “displacement score” = 2 × (C − D)/(A + B), where A is the longest diameter of the placenta, B is the shortest diameter of the placenta, C is the longest distance from CI to the margin of the placenta, and D is the shortest distance from CI to the margin of the placenta.

Using this formula, the score will decrease where the CI is near the center of the placenta; on the other hand, the score will approach 1 where the CI is located near the margins of the placenta. It becomes greater than 1 if the CI attaches directly to the membrane away from the placental margin.

Figure 2 shows the actual placenta and measurement sites of umbilical CI. Images (i), (ii), and (iii) indicate a normal CI, a marginal CI (MCI), and a VCI, respectively. Image (ii) is a case of MCI where D is zero. Furthermore, images (iv), (v), and (vi) are the same case and VCI with branched blood vessels on the velamentum. Image (iv) and (vi) are pictures with the umbilical cord pulled to the opposite side, and (v) is an image with the umbilical cord lifted from the placenta. In such cases, CI is the site where Wharton’s jelly disappeared, and D is the shortest distance to the placental margin.

MCI was defined as CI with Wharton’s jelly attached 2 cm or less from the margin of the placenta, and VCI was defined as vulnerable umbilical vessels without Wharton’s jelly running through the fetal membrane [7].

For the comparison of birth weight, an examination between the two groups was performed by comparing the difference in the standard deviation (SD) between the two groups in terms of how much their birth weight deviated from the mean of their birth week. Birth weight SD was calculated using the “Japanese Birth Weight SD Score Table” based on the number of weeks of delivery, nulligravida/multigravida, sex of the infant, and birth weight [8]. Furthermore, the hypertensive disorders of pregnancy (HDP) of each patient were diagnosed based on the diagnostic criteria of the Japanese Society of Gestational Hypertension [9].

Median comparisons between the two groups were made using the Mann–Whitney U test, and median comparisons between the three groups were made using the Kruskal–Wallis test. Categorical data were expressed as frequencies and percentages, and the Pearson χ-square test was used for these variables. Linear regression models were used for multivariate analysis of continuous variables. Logistic regression models were used for multivariate analyses of binary variables, and R version 4.2.2, EZR version 1.55, and a Microsoft Excel (2018) spreadsheet were used for statistical analyses. Statistical significance was defined as a two-sided *p*-value < 0.05.

## 3. Results

Figure 3 shows the inclusion and exclusion criteria of this study. During the study period, 1426 cases were delivered at our hospital, and 324 of these cases were excluded. Of the 1102 singleton pregnancies, 934 were spontaneously conceived, and 168 were ART pregnancies. Among the ART cases, 104 were cIVF, and 64 were IVF/ICSI.

Table 1 summarizes and compares the characteristics of the subjects. The maternal age was significantly higher in the ART group (38 vs. 32 years, *p* < 0.001). Moreover, the proportion of multiparous women was significantly higher in the spontaneous conception group than that in the ART group (405/934: 43.3% vs. 51/168: 30.4%, *p* < 0.01).

Table 2 summarizes and compares the demographic characteristics related to the perinatal outcome in the study. The ART group had a significantly higher rate of emergency cesarean section than the spontaneous conception group (48/168: 28.6% vs. 179/934: 19.2%, *p* < 0.01). Blood loss at delivery in the ART group was significantly more frequent (750 g vs. 445 g, *p* < 0.001), and the SD of birth weight was also significantly higher (0.265 vs. 0.041, *p* < 0.05). Moreover, the rate of VCI in the ART group was significantly higher (11/168: 6.5% vs. 15/934: 1.6%, *p* < 0.001), and the rate of VP was also significantly higher (11/168: 6.5% vs. 13/934: 1.4%, *p* < 0.001). The rate of placenta previa in the ART group was found to be significantly higher as well (13/168: 7.7% vs. 37/934: 4.0%, *p* < 0.05).

Table 3 shows the results for the SD of birth weight, adjusted for background factors. The SD of infant birth weight was higher in cIVF and IVF/ICSI, compared with spontaneous conception (regression coefficient estimates 0.258/0.335, 95% confidence interval 0.012–0.506, *p* < 0.05/0.032–0.639, *p* < 0.05, respectively).

Table 4 presents the odds ratios for the rate of gestational hypertension, also adjusted for background factors. The results show that the odds ratios for gestational hypertension in pregnancies due to cIVF and IVF/ICSI were 1.773 (95% confidence interval: 0.779–4.034. *p* = 0.172) and 1.919 (95% confidence interval: 0.648–5.684, *p* = 0.239), respectively, compared with spontaneous conception. There were no statistically significant differences in this study.

Table 5 shows a comparison of placental weight, VCI, MCI, VP, placenta previa, and low-lying placenta among the three groups. VCI was significantly higher in the cIVF and IVF/ICSI groups than in the spontaneous group; however, there was no difference in MCI among the three groups.

Table 6 shows the results of a multivariate analysis of the displacement scores among the three groups of spontaneous conception, cIVF, and IVF/ICSI. The displacement score was significantly higher in the cIVF and IVF/ICSI groups than in the spontaneous conception group, and the displacement score was also significantly higher in the IVF/ICSI group than in the cIVF group. Table 7 shows the result of higher displacement scores in the IVF/ICSI group in normal CI and MCI cases (total of 1076 cases); even in cases excluding VCI, the displacement scores were higher in the IVF/ICSI group than in the spontaneous conception group.

## 4. Discussion

The present study first demonstrated that IVF/ICSI has a striking impact on the umbilical insertion site into the placenta when compared with placentas from spontaneously conceived pregnancies. Furthermore, we also quantitatively clarified that the significant deviations of CI on the placental surface may be caused by IVF/ICSI, even in the placentas without VCI.

Clinical features of placentas and CI regarding the mode of conception have been reported in previous studies [6,10]. Englert et al. first demonstrated a significantly higher frequency of MCI (15%) and VCI (14%) in cIVF groups when compared with a general obstetrical population (MCI, 6% and VCI, 1%, respectively), although there was no difference in the placental morphology between the two groups [10]. Soon after that, Jauniaux et al. reported a significant difference in placental shapes and CI between a group of placentas collected from pregnancies resulting from cIVF and a control group composed of placentas from spontaneous pregnancies [6]. In their study, MCI (26%) and VCI (12%) in the cIVF group were more frequently observed when compared with the control groups (MCI, 10% and VCI, 2%, respectively). When compared with these previous reports, however, our present study showed discrepancies in the number of frequencies. MCI was observed in 15.5% of the ART group and 5.5% of the control group. Moreover, VCI was observed in 6.5% of the ART group and 1.6% of the control group. Comparing our results with those from previous research, the most significant difference in the procedural background might be that the method of conception was fresh embryo transfer in previous reports and frozen embryo transfer in our study, and this might be one of the explanations for the discrepancies among the studies.

Jauniaux et al. speculated in their previous study that the artificial placement of the embryo in the uterine cavity could impede the normal process of blastocyst orientation during its implantation period, which might thus cause VCI [6]. Blastocysts already have a sufficient capacity to be implanted into an optimally synchronized endometrium; in contrast, cleavage-stage embryos need more time (2–3 days) until the implantation process is initiated. These chronological differences between a cleavage-stage embryo and a blastocyst-stage embryo may reflect the difference in the incidence of VCI [7]. Monie et al. also reported a study of the pathogenesis of VCI [11]. Hasegawa et al. have further developed the theory of Monie et al. that placental adhesion in the lower uterine segment may be involved in the etiology of VCI [12]. Their study reported that VCI is common in cases in which the placenta is attached to the lower uterine segment, and they also showed that the location of the umbilical CI may be determined at the first process when the trophoblast cells of the embryo implant into the lower uterine segment and proliferate and invade the villous cavity. The subsequent progression of placental migration toward the fundus of the uterus may further result in locating the umbilical CI far away from the center of the placenta. However, over the past three decades, these hypotheses have never been proven, no doubt due to the complexity of the implantation phenomenon.

Our study showed that VCI was more commonly observed in cases conceived by IVF/ICSI. The procedures for IVF/ICSI involve removing the cumulus cells around the retrieved oocyte, puncturing the zona pellucida, and then injecting the sperm directly into the cytoplasm in order to inseminate the oocyte [13]. These technical differences, especially on the zona pellucida, between cIVF and IVF/ICSI may contribute to subsequent placentation and variation in the umbilical CI site.

The zona pellucida prevents multisperm fertilization and protects the early embryo prior to implantation, and it is composed of oocytes and a small number of egg-specific glycoproteins zona pellucida 1–4, which, like other somatic extracellular matrix (ECM) cells, are thought to influence cell adhesion and migration, intercellular communication, gene expression, differentiation, and morphogenesis [14]. Regarding its roles in detail, the zona pellucida assists in the development of oocytes and follicles during oogenesis, expresses species-specific receptors for sperm to bind to the egg during fertilization, and causes physical and immunological changes that help prevent polyspermy after fertilization [15]. In 1989, Malter et al. reported on the relationship between assisted hatching (AH) and VCI in terms of manipulation of the transparent zone and placental and umbilical cord formation [16]. They speculated that AH may cause abnormal placental and umbilical cord formation because it penetrates the endometrium more quickly than normal. They further reported that inadequate AH can impede complete hatching during the incubation process, therefore causing problems with subsequent embryonic development, which can lead to placental malformation, miscarriage, or twin pregnancies [16]. Herman et al. also reported that since hatching is performed over the trophectoderm cells, which will subsequently develop into the placenta, the act of AH may be hypothesized to affect these cells and thus lead to increased rates of bilobated placentas [17].

Issues on the relationship between IVF/ICSI and embryo hatching have been discussed previously [18]. Regarding the process of hatching, fluid is taken into the blastocyst as the embryo matures, therefore causing an increase in hydrostatic pressure and stretching of the trophoblastic epithelium [18]. Then, through cell proliferation and proper expansion of the blastocyst, the volume of the trophoblast cells increases, the zona pellucida melts and thins, and hatching occurs [19]. Inoue et al. reported that in incompletely hatched IVF/ICSI blastocysts, the trophectoderm (TE) and inner cell mass (ICM) cells herniate and degenerate intermittently through small slits [18]. They speculate that this may be due to the fact that the small slits caused by the procedure of sperm injection in the zona pellucida allow fluid to flow out of the blastocyst and the hydrostatic pressure does not increase, thus obstructing the hatching process [18]. Furthermore, IVF/ICSI lacks the process of a zona pellucida–oocyte acrosome reaction, which generates functional changes in the nature of the zona pellucida [20,21,22]. These facts may also explain the influences of IVF/ICSI procedures on the subsequent hatching and implantation process.

In terms of birth weight, our study confirms a trend toward larger birth weights with ART compared with spontaneous conception. Many papers have reported on the birth weight of infants born via ART versus spontaneous conception; however, there are differences in the timing of the reports, and there are no consistent views. A previous study of over 2 million births reported a 13% increase in the frequency of small for gestational age (SGA) children born via ART (odds ratio 1.13; 95% CI, 1.10–1.15) [23]. Qin et al. also noted an increased risk of SGA in pregnancies through ART in a large meta-analysis (odds ratio 1.35; 95% CI 1.20–1.52) [24]. However, recent papers have reported that ART does not affect the birth of SGA infants but, on the contrary, increases their larger weights. In a 2018 report, Hwang et al. stated that the risk of SGA was reduced in their ART group [25]. In addition, Glatthorn et al. also reported a lower risk of SGA, compared with spontaneous conception, based on data from 16 million people from 2015 to 2019. They suggest that their observation is related to changes in maternal lifestyle, maternal compliance of infertility patients, and other factors compared with previous years [26]. These observations are in line with our findings.

Our study has some limitations. The first one is the relatively small number of subjects. The second is that our study lacks a comparison of the outcome depending on the mode of conception, such as fresh or frozen embryo transfer. The last one is that the present study did not contain a detailed consideration of the development of ART devices and techniques over the last few decades.

## 5. Conclusions

Our study first demonstrated that the umbilical CI on the placental surface could be affected by the methods of ART; furthermore, it clearly showed that IVF/ICSI might further displace the CI from the center of the placenta. With these findings, close information sharing between fertility practitioners and obstetricians, as well as careful observation of the fetus and the placenta in early pregnancy, will improve the prognosis for fetuses and newborns from disastrous outcomes caused by cord troubles, especially vasa previa. In addition, by obtaining prior information on VCI, obstetricians will be more vigilant in the management of pregnant women in labor or with rupture of the membranes, when attempting a vaginal delivery. Further research should be undertaken to determine whether fresh embryo transfer or frozen embryo transfer has a greater impact on the umbilical CI site, considering the physiology of the zona pellucida and the remarkable progress of reproductive techniques.

## Figures and Tables

**Figure 1 medicina-59-01715-f001:**
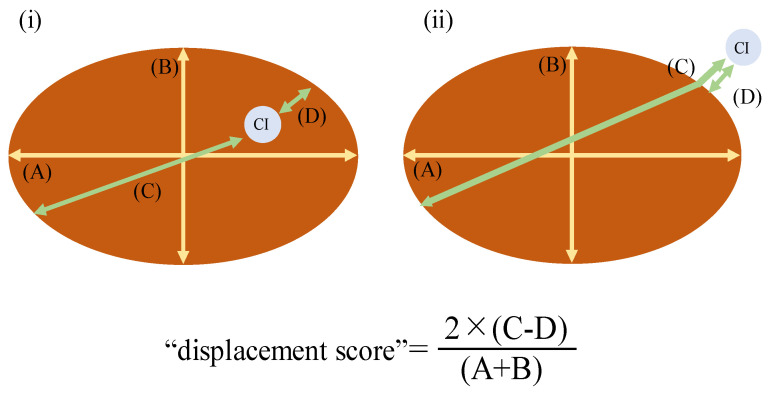
How to measure the umbilical cord insertion (CI). This figure shows the “displacement score” measurement method. (A): the longest diameter of the placenta, (B): the shortest diameter of the placenta, (C): the longest distance from CI to the margin of the placenta, (D): the shortest distance from CI to the margin of the placenta, CI: cord insertion (**i**) Normal CI, (**ii**) VCI.

**Figure 2 medicina-59-01715-f002:**
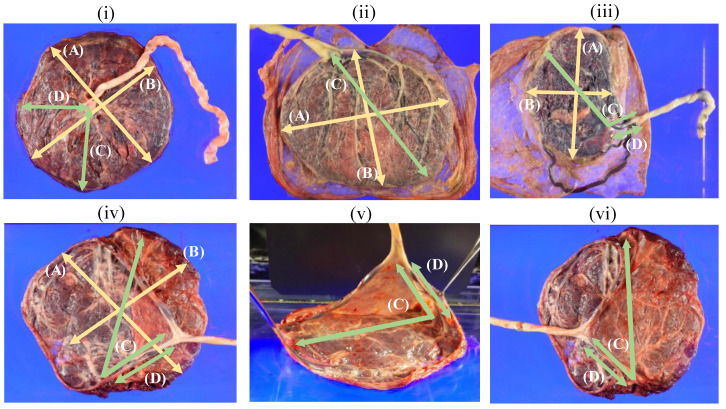
The actual placenta and measurement sites of umbilical CI. (A): the longest diameter of the placenta, (B): the shortest diameter of the placenta, (C): the longest distance from CI to the margin of the placenta, (D): the shortest distance from CI to the margin of the placenta. The pictures of (**i**), (**ii**), and (**iii**) indicate normal CI, MCI, and VCI, respectively. The picture of (**ii**) is a case of MCI and D is zero. The pictures of (**iv**–**vi**) are VCI with branched blood vessels on the velamentum in the same case. The pictures (**iv**,**vi**) show the umbilical cord pulled to the opposite side, and (**v**) is a picture with the umbilical cord lifted from the placenta. In such cases, CI is the site where Wharton’s jelly disappeared, and D is the shortest distance to the placental margin.

**Figure 3 medicina-59-01715-f003:**
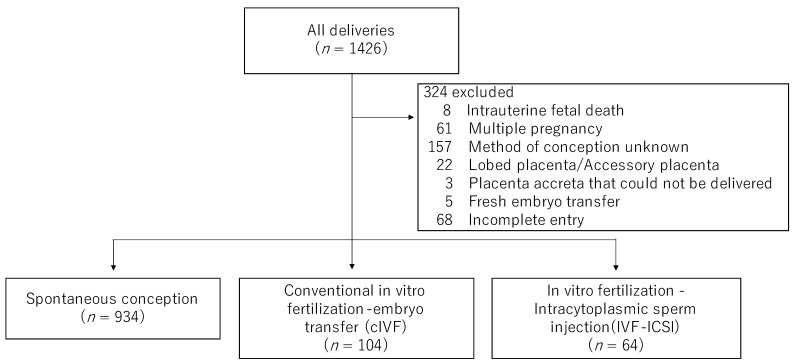
The inclusion and exclusion criteria of this study. During the study period, 1426 cases were delivered at our hospital, and 324 cases were excluded. Of the 1102 singleton pregnancies, 934 were spontaneously conceived and 168 were ART pregnancies. In the ART cases, 104 were cIVF, and 64 were IVF/ICSI.

**Table 1 medicina-59-01715-t001:** Characteristics.

	Spontaneous Conception Group (n = 934)	Assisted Reproductive Technology Pregnancy Group (n = 168)	*p*-Value
Age (years) (range)	32 (17–45)	38 (26–49)	<0.001 ^a^
Gravidity			0.536 ^b^
Nulligravida	382	73	
Multigravida (range)	552 (2–10)	95 (2–7)	
Parity			0.002 ^b^
Nullipara	529	117	
Multipara (range)	405 (1–6)	51 (1–3)	
Prepregnancy BMI (kg/m^2^) (range)	21.0 (15.0–42.9)	20.5 (17.3–37.0)	0.086 ^a^
Hypothyroidism	56 (6.0%)	40 (23.8%)	<0.001 ^b^
Hyperthyroidism	22 (2.4%)	6 (3.6%)	0.356 ^b^
Type 1 diabetes mellitus	10 (1.1%)	1 (0.6%)	0.568 ^b^
Type 2 diabetes mellitus	12 (1.3%)	1 (0.6%)	0.446 ^b^
Uterine myoma	68 (7.3%)	19 (11.3%)	0.075 ^b^
Uterine adenomyosis	7 (0.7%)	4 (2.4%)	0.050 ^b^
Asthma	88 (9.4%)	16 (9.5%)	0.967 ^b^
Epilepsy	24 (2.6%)	2 (1.2%)	0.278 ^b^
Schizophrenia	6 (0.6%)	1 (0.6%)	0.944 ^b^
Depression	22 (2.4%)	2 (1.2%)	0.341 ^b^
Systemic lupus erythematosus	2 (0.2%)	1 (0.6%)	0.383 ^b^
Idiopathic thrombocytopenic purpura	9 (1.0%)	2 (1.2%)	0.785 ^b^

^a^ Mann–Whitney U test; ^b^ Chi-square.

**Table 2 medicina-59-01715-t002:** Perinatal outcome.

	Spontaneous Conception Group (n = 934)	Assisted Reproductive Technology Pregnancy Group (n = 168)	*p*-Value
BMI at delivery (kg/m^2^) (range)	24.9 (17.9 to 44.2)	24.5 (17.2 to 36.9)	0.008 ^a^
Weight gain during pregnancy (kg) (range)	9.9 (−12.0 to 28.8)	9.6 (−3.8 to 17.0)	0.125 ^a^
Gestational age (weeks)	38.7 (22.3 to 41.7)	38.6(27.9 to 41.6)	0.935 ^a^
Emergency cesarean section	179 (19.2%)	48 (28.6%)	0.007 ^b^
Nonreassuring fetal status	57 (31.8%)	11 (22.9%)	0.288 ^b^
Cessation of labor	53 (29.6%)	17 (35.4%)	0.483 ^b^
Hypertensive disorders of pregnancy	14 (7.8%)	7 (14.6%)	0.164 ^b^
Instrumental delivery	47 (5.0%)	12 (7.1%)	0.473 ^b^
Nonreassuring fetal status	37 (78.7%)	8 (66.7%)	0.453 ^b^
Hypertensive disorders of pregnancy	3 (6.4%)	2 (16.7%)	0.266 ^b^
Blood loss at delivery (g) (range)	445 (25 to 4280)	750 (85 to 3170)	<0.001 ^a^
Hypertensive disorders of pregnancy	68 (7.3%)	19 (11.3%)	0.075 ^b^
Gestational Diabetes Mellitus	99 (10.6%)	22 (13.1%)	0.341 ^b^
Birth weight (g)	2915 (257 to 4455)	2965 (767 to 4420)	0.146 ^a^
Birth weight (SD)	0.041 (−5.058 to 3.970)	0.265 (−2.913 to 4.453)	0.019 ^a^
Light for dates infant	72 (7.7%)	15 (8.9%)	0.589 ^b^
Heavy for dates infant	82 (8.8%)	19 (11.3%)	0.295 ^b^
Umbilical cord length (cm)	52 (17 to 104)	53 (28 to 98.5)	0.621 ^a^
Placental weight (g) (range)	550 (110 to 1120)	555 (225 to 1040)	0.212 ^a^
Velamentous cord insertion	15 (1.6%)	11 (6.5%)	<0.001 ^b^
Marginal cord insertion	51 (5.5%)	12 (7.1%)	0.387 ^b^
Vasa previa	13 (1.4%)	11 (6.5%)	<0.001 ^b^
Placenta previa	37 (4.0%)	13 (7.7%)	0.030 ^b^
Low-lying placenta	25 (2.7%)	6 (3.6%)	0.518 ^b^
Placenta accreta	3 (0.3%)	1 (0.6%)	0.587 ^b^
Smoking during pregnancy	21 (2.2%)	0 (0.0%)	0.0553 ^b^

^a^ Mann-Whitney U test; ^b^ Chi-square.

**Table 3 medicina-59-01715-t003:** Multivariate analysis of birth weight (SD).

	Estimates ^a^	95% CI ^b^	*p*-Value
cIVF (ref = spontaneous conception)	0.258	0.012 to 0.506	0.040
IVF/ICSI (ref = spontaneous conception)	0.335	0.032 to 0.639	0.031
IVF/ICSI (ref = cIVF)	0.077	−0.283 to 0.437	0.674

^a^ Adjusted for gestational diabetes mellitus, hypertensive disorders of pregnancy, hypothyroidism, hyperthyroidism, multipara, weight gain during pregnancy (kg), age (years), BMI at delivery (kg/m^2^), smoking during pregnancy; ^b^ 95% confidence intervals were estimated using linear regression analysis. cIVF = conventional in vitro fertilization. IVF/ICSI = in vitro fertilization/intracytoplasmic sperm injection.

**Table 4 medicina-59-01715-t004:** Multivariate analysis of hypertensive disorders of pregnancy.

	Odds Ratio ^a^	95% CI ^b^	*p*-Value
cIVF (ref = spontaneous conception)	1.773	0.779–4.034	0.172
IVF/ICSI (ref = spontaneous conception)	1.919	0.648–5.684	0.239
IVF/ICSI (ref = cIVF)	1.083	0.324 to 3.615	0.897

^a^ Adjusted for gestational diabetes mellitus, hypothyroidism, hyperthyroidism, birth weight (SD), non-pregnant BMI (kg/m^2^), weight gain during pregnancy (kg), age (years), smoking during pregnancy, multipara; ^b^ 95% confidence intervals were estimated using logistic regression analysis. cIVF = conventional in vitro fertilization. IVF/ICSI = in vitro fertilization/intracytoplasmic sperm injection.

**Table 5 medicina-59-01715-t005:** Perinatal outcomes related to placenta and umbilical cord.

	Spontaneous Conception Group (n = 934)	cIVF Group (n = 104)	IVF/ICSI Group (n = 64)	*p*-Value
Placental weight (g) (range)	550 (110–1120)	555 (225–905)	555 (280–1040)	0.414 ^a^
Velamentous cord insertion	15 (1.6%)	5 (4.8%)	6 (9.4%)	<0.001 ^b^*^1^
Marginal cord insertion	51 (5.5%)	7 (6.7%)	5 (7.8%)	0.531 ^b^
Vasa previa	13 (1.4%)	5 (4.8%)	6 (9.4%)	<0.001 ^b^*^2^
Placenta previa	37 (4.0%)	10 (9.6%)	3 (4.7%)	0.040 ^b^*^3^
Low-lying placenta	25 (2.7%)	1 (1.0%)	5 (7.8%)	0.041 ^b^*^4^

^a^ Kruskal–Wallis test; ^b^ Chi-square. *^1^ Spontaneous conception vs. cIVF *p* = 0.042, IVF/ICSI vs. spontaneous conception *p* = 0.001; *^2^ Spontaneous conception vs. cIVF *p* = 0.027, IVF/ICSI vs. spontaneous conception *p* < 0.001; *^3^ Spontaneous conception vs. cIVF *p* = 0.021; *^4^ cIVF vs. IVF/ICSI *p* = 0.030, IVF/ICSI vs. spontaneous conception *p* = 0.038.

**Table 6 medicina-59-01715-t006:** Multivariate analysis of displacement scores (including VCI).

	Estimates ^a^	95% CI ^b^	*p*-Value
cIVF (ref = spontaneous conception)	0.041	−0.021 to 0.104	0.196
IVF/ICSI (ref = spontaneous conception)	0.201	0.124 to 0.278	<0.001
IVF/ICSI (ref = cIVF)	0.159	0.069 to 0.250	0.001

^a^ Adjusted for gestational diabetes mellitus, hypertensive disorders of pregnancy, hypothyroidism, hyperthyroidism, multipara, weight gain during pregnancy (kg), birth weight (SD), female, age (years), nonpregnant BMI (kg/m^2^), smoking during pregnancy; ^b^ 95% confidence intervals were estimated using linear regression analysis. cIVF = conventional in vitro fertilization. IVF/ICSI = in vitro fertilization/intracytoplasmic sperm injection. VCI = velamentous cord insertion.

**Table 7 medicina-59-01715-t007:** Multivariate analysis of displacement scores (excluded VCI).

	Estimates ^a^	95% CI ^b^	*p*-Value
cIVF (ref = spontaneous conception)	0.017	−0.043 to 0.076	0.581
IVF/ICSI (ref = spontaneous conception)	0.104	0.028 to 0.181	0.008
IVF/ICSI (ref = cIVF)	0.088	−0.002 to 0.177	0.055

^a^ Adjusted for gestational diabetes mellitus, hypertensive disorders of pregnancy, hypothyroidism, hyperthyroidism, multipara, weight gain during pregnancy (kg), birth weight (SD), female, age (years), nonpregnant BMI (kg/m^2^), smoking during pregnancy; ^b^ 95% confidence intervals were estimated using linear regression analysis. cIVF = conventional in vitro fertilization. IVF/ICSI = in vitro fertilization/intracytoplasmic sperm injection. VCI = velamentous cord insertion.

## Data Availability

The datasets used and/or analyzed during the current study are available from the corresponding author upon reasonable request.

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
