# Peer review of "The Impact of Assisted Reproductive Technology on Umbilical Cord Insertion: Increased Risk of Velamentous Cord Insertion in Singleton Pregnancies Conceived through ICSI"

_medicina, 2023, doi:10.3390/medicina59101715_

Round 1

Reviewer 1 Report

This is a monocentric retrospective, observational study which focuses on a specific correlation between mode of conception and umbilical cord implantation.

There are several positive aspects: the originality of the research and of its findings, there is a good selection of the cases and the statistical study is well conducted.

The conclusions should remain on the objects of the study. The digression on the histopathological aspects and the possible mechanisms of different implantations are too detailed and often not connected with the object of the research. It creates a serious decrease of interest in the reader and should be cut. Besides, the very detailed description of the differences between the two groups transcends the aims of the study and gives an unnecessary burden.

Instead, it should be better explained how the conclusions could have an impact on the clinical management of these pregnancies.

Reviewer 2 Report

In the manuscript entitled “The impact of ICSI on umbilical cord insertion: Why should we know the mode of conception in perinatal surveillance?” , Fukuda et al performed a retrospective study where they evaluated the deviations of umbilical cord insertions from the center of the placenta, and depicted the association of conventional In vitro fertilization (cIVF) and In vitro fertilization-intracytoplasmic sperm injection (IVF-ICSI) with velamentous cord insertion  (VCI). In this retrospective study a total of 1,102 patients who delivered after 22 weeks of gestation were included and were divided into three groups: spontaneous pregnancy, cIVF, and IVF-ICSI. Various data such as background information, perinatal complications, perinatal outcomes, and a numerical “displacement score” were collected and accordingly analysis was done. The authors observed that as compared with spontaneous conception group the displacement score was significantly higher in the cIVF and IVF-ICSI groups; and amongst cIVF and IVF-ICSI group, IVF-ICSI group demonstrated a significantly higher displacement score. Overall, their study suggested the involvement of ART  technique in influencing the location of umbilical CI on the surface of placenta where IVF-ICSI poses a higher risk for displacement of CI from the placental center. The findings of the present study will help the fertility practitioners and obstetricians in proper management of such cases by careful observation of the fetus and placenta during early pregnancy in such cases using ART techniques and thus improving the outcome. Overall, the study is well planned and results are well presented.  Discussion and conclusion are appropriate. The manuscript is clearly written. I just have a couple of suggestions that may improve the manuscript.

1. The tile can be improved.

2. The authors can explain Vasa previa (VP) and velamentous cord insertion (VCI) more clearly to improve the readership.

3. Some of the old references may be replaced with recent relevant ones.

Well written. 
